# The mirror mode: A "superconducting" space plasma analogue

Rudolf A. Treumann[1] and Wolfgang Baumjohann[2]

[1]International Space Science Institute, Bern, Switzerland
[2]Space Research Institute, Austrian Academy of Sciences, Graz, Austria,
*Correspondence to*: Wolfgang.Baumjohann@oeaw.ac.at

*Abstract.*– We examine the physics of the magnetic mirror mode in its final state of saturation, the thermodynamic equilibrium, to demonstrate that the mirror mode is the analogue of a superconducting effect in a classical anisotropic-pressure space plasma. Two different spatial scales are identified which control the behaviour of its evolution. These are the ion inertial scale $\lambda_{im}(\tau)$ based on the excess density $N_m(\tau)$ generated in the mirror mode, and the Debye scale $\lambda_D(\tau)$. The Debye length plays the role of the correlation length in superconductivity. Their dependence on the temperature ratio $\tau = T_\parallel/T_\perp < 1$ is given, with $T_\perp$ the reference temperature at critical magnetic field. The mirror mode equilibrium structure under saturation is determined by the Landau-Ginzburg ratio $\kappa_D = \lambda_{im}/\lambda_D$, or $\kappa_\rho = \lambda_{im}/\rho$, depending on whether the Debye length or the thermal-ion gyroradius $\rho$ – or possibly also an undefined turbulent correlation length $\ell_{turb}$ – serve as correlation lengths. Since in all space plasmas $\kappa_D \gg 1$, plasmas with $\lambda_D$ as relevant correlation length always behave like type II superconductors, naturally giving rise to chains of local depletions of the magnetic field of the kind observed in the mirror mode. In this way they would provide the plasma with a short scale magnetic bubble texture. The problem becomes more subtle when $\rho$ is taken as correlation length. In this case the evolution of mirror modes is more restricted. Their existence as chains or trains of larger scale mirror bubbles implies that another threshold, $V_A > v_{\perp th}$, is exceeded. Finally, in case that the correlation length $\ell_{turb}$ instead results from low frequency magnetic/magnetohydrodynamic turbulence, the observation of mirror bubbles and the measurement of their spatial scales sets an upper limit on the turbulent correlation length. This might be important in the study of magnetic turbulence in plasmas.

## 1 Introduction

Under special conditions high-temperature collisionless plasmas may develop properties which resemble those of superconductors. This is the case with the mirror mode when the anisotropic pressure gives rise to local depletions of the magnetic field similar to the Meissner effect in metals where it signals the onset of superconductivity (Kittel, 1963; Fetter & Walecka, 1971; Huang, 1987; Lifshitz & Pitaevskii, 1998), i.e. the suppression of friction between the current and the lattice. In collisionless plasmas there is no lattice, the plasma is frictionless, thus it already is ideally conducting which, however, does not mean that it is superconducting! For being superconducting, additional properties are required. These, as we show below, are given in the saturation state of the mirror mode.

The mirror mode is a non-oscillatory plasma instability (Chandrasekhar, 1961; Hasegawa, 1969; Gary, 1993; Southwood & Kivelson, 1993; Kivelson & Southwood, 1996) which evolves in anisotropic plasmas (for a recent review see Sulem, 2011,

and references therein). It has been argued that it should readily saturate by quasilinear depletion of the temperature anisotropy (cf., e.g. Noreen et al., 2017, and references therein). Observations do not support this conclusion. In fact, we recently argued (Treumann & Baumjohann, 2018a) that the large amplitudes of mirror-mode oscillations observed in the Earth's magnetosheath, magnetotail and elsewhere, like other planetary magnetosheaths, in the solar wind and generally in the heliosphere,

(see., e.g. Tsurutani et al., 1982, 2011; Czaykowska et al., 1998; Zhang et al., 1998; Constantinescu et al., 2003; Zhang et al., 2008, 2009; Lucek et al., 1999a, b; Volwerk et al., 2008, and many others) are a sign of the impotence of quasilinear theory of limiting the growth of the mirror instability. Instead, mirror modes should be subject to weak kinetic turbulence theory (Sagdeev & Galeev, 1969; Davidson, 1972; Tsytovich, 1977; Yoon, 2007, 2018; Yoon & Fang, 2007) which allows them to evolve until becoming comparable in amplitude to the ambient magnetic field long before any dissipation can set on.

This is not unreasonable, because all those plasmas where the mirror instability evolves are ideal conductors on the scales of the plasma flow. On the other hand, no weak turbulence theory of the mirror mode is available yet as it is difficult to identify the various modes which interact to destroy quasilinear quenching. The frequent claim that whistlers (lion roars) excited in the trapped electron component would destroy the bulk (global) temperature anisotropy is erroneous, because whistlers (Thorne & Tsurutani, 1981; Baumjohann et al., 1999; Maksimovic et al., 2001; Zhang et al., 1998) grow on the expense of a small

component of anisotropic resonant particles only (Kennel & Petschek, 1966). Depletion of the resonant anisotropy does by no means affect the bulk temperature anisotropy that is responsible for the evolution of the mirror instability. On the other hand, construction of a weak turbulence theory of the mirror mode poses serious problems. One therefore needs to refer to other means of description of its final saturation state.

  Since measurements suggest that the observed mirror modes are about stationary phenomena which are swept over the space-

craft at high flow speeds (called Taylor's hypothesis though, in principle, just refers to the Galilei transformation), it seems reasonable to tackle them within a *thermodynamic* approach, i.e. assuming that in the observed large amplitude saturation state they can be described as the *stationary* state of interaction between the ideally conducting plasma and magnetic field. This can be most efficiently done when the free energy of the plasma is known which, unfortunately, is not the case. Magnetohydrodynamics does not apply, and the formulation of a free energy in the kinetic state is not available. For this reason we refer to some

phenomenological approach which is guided by the phenomenological theory of superconductivity. There we have the similar phenomenon that the magnetic field is expelled from the medium due to internal quantum interactions, known as the Meissner effect. This resembles the evolution of the mirror mode though in our case the interactions are not in the quantum domain. This is easily understood if considering the thermal length $\lambda_\hbar = \sqrt{2\pi\hbar^2/m_e T}$ and comparing it to the shortest plasma scale, viz. the inter-particle distance $d_N \sim N^{-\frac{1}{3}}$. The former is, for all plasma temperatures $T$, in the atomic range while the latter in

space plasmas for all densities $N$ is at least several orders of magnitude larger. Plasmas are classical. In their equilibrium state classical thermodynamics applies to them.

  In the following we boldly ask for the *thermodynamic* equilibrium state of a mirror unstable plasma. (For other non-thermodynamical attempts of modelling the equilibrium configuration of magnetic mirror modes and application to multi-spacecraft observations, the reader may consult Constantinescu, 2002; Constantinescu et al., 2003). Such an approach is rather

alien to space physics. It follows the path prescribed in solid state physics but restricts itself to the domain of classical thermodynamics only.

## 2 Properties of the mirror instability

The mirror instability evolves whence the magnetic field $B$ in a collisionless magnetised plasma with an internal pressure/temperature anisotropy $T_\perp > T_\parallel$, where the subscripts refer to the directions perpendicular and parallel to the ambient magnetic field, drops below a critical value

$$B < B_{crit} \approx \sqrt{2\mu_0 N T_{i\perp}} \left( \Theta_i + \sqrt{\frac{T_{e\perp}}{T_{i\perp}}} \Theta_e \right)^{\frac{1}{2}} \left| \sin \theta \right| \tag{1}$$

where $\Theta_j = \left( T_\perp / T_\parallel - 1 \right)_j > 0$ is the temperature anisotropy of species $j = e, i$ (for ions and electrons) and $\theta$ is the angle of propagation of the wave with respect to the ambient magnetic field (cf., e.g., Treumann & Baumjohann, 2018a). Here any possible temperature anisotropy in the electron population has been included but will be dropped below as it seems (Yoon & López, 2017) that it does not provide any further insight into the physics of the final state of the mirror mode.

The important observation is that the existence of the mirror mode depends on the temperature difference $T_\perp - T_\parallel$ and the critical magnetic field. Commonly only the temperature anisotropy is reclaimed as being responsible for the growth of the mirror mode. Though this is true, it also implies the above condition on the magnetic field. To some degree this resembles the behaviour of magnetic fields under superconducting conditions. To demonstrate this, we take $T_\perp$ as reference – or critical – temperature. The critical magnetic field becomes a function of the temperature ratio $\tau = T_\parallel / T_\perp$. Once $\tau < 1$ and $B < B_{crit}$ the magnetic field will be pushed out of the plasma to give space to an accumulated plasma density and also weak diamagnetic surface currents on the boundaries of the (partially) field-evacuated domain.

The $\tau$-dependence of the critical magnetic field can be cast into the form

$$\frac{B_{crit}(T_\parallel)}{B_{crit}^0} = \left[ \tau^{-1} \left( 1 - \tau \right) \right]^{\frac{1}{2}} = \left( \frac{T_\perp}{T_\parallel} \right)^{\frac{1}{2}} \left( 1 - \frac{T_\parallel}{T_\perp} \right)^{\frac{1}{2}} \tag{2}$$

which indeed resembles that in the phenomenological theory of superconductivity. Here

$$B_{crit}^0 = \sqrt{2\mu_0 N T_{i\perp}} \left| \sin \theta \right| \tag{3}$$

and the critical threshold vanishes for $\tau = 1$ where the range of possible unstable magnetic field values shrinks to zero; the limits $T_\parallel = 0$ or $T_\perp = \infty$ make no physical sense.

Though the effects are similar to superconductivity, the temperature dependence is different from that of the Meissner effect in metals in their isotropic low-temperature super-conducting phase. In contrast, in an anisotropic plasma the effect occurs in the high-temperature phase only while being absent at low temperatures. Nevertheless, the condition $\tau < 1$ indicates that the mirror mode, concerning the ratio of parallel to perpendicular temperatures, is a *low-temperature* effect in the high-temperature plasma phase. This may suggest that even in metals high-temperature superconductivity might be achieved more easily for anisotropic temperatures, a point we will follow elsewhere (Treumann & Baumjohann, 2018b).

Since the plasma is ideally conducting, any quasi-stationary magnetic field is subject to the penetration depth, which is the inertial scale $\lambda_{im} = c/\omega_{im}$, with $\omega_{im}^2 = e^2 N_m/\epsilon_0 m_i$ based on the density $N_m$ of the plasma component involved into the mirror effect. The mirror instability is a slow purely growing instability with real frequency $\omega \approx 0$. On these low frequencies the plasma is quasi-neutral. In metallic superconductivity this length is the London penetration depth which refers to electrons as the ions are fixed to the lattice. Here, in the space plasma, it is rather the ion scale because the dominant mirror effect is caused by mobile ions in the absence of any crystal lattice. Such a "magnetic lattice" structure is ultimately provided under conditions investigated below by the saturated state of the mirror mode, where it collectively affects the trapped ion component on scales of an internal correlation length.

## 3 Free energy

In the thermodynamic equilibrium state the quantity which describes the matter in the presence of a magnetic field $\boldsymbol{B}$ is the Landau-Gibbs free energy density

$$G_L = F_L - \frac{1}{2\mu_0}\delta\boldsymbol{B}\cdot\boldsymbol{B} \tag{4}$$

where $F_L$ is the Landau free energy density (Kittel & Kroemer, 1980) which, unfortunately, is not known. In magnetohydrodynamics it can be formulated but becomes a messy expression which contains all stationary, i.e. time-averaged, nonlinear contributions of low-frequency electromagnetic plasma waves and thermal fluctuations. The total Landau-Gibbs free energy is the volume integral of this quantity over all space. In thermodynamic equilibrium this is stationary, and one has

$$\frac{d}{dt}\int d^3x \, G_L = 0 \tag{5}$$

In order to restrict to our case we assume that $F_L$ in the above expression, which contains the full dynamics of the plasma matter, can be expanded with respect to the normalised density $N_m < 1$ of the plasma component which participates in the mirror instability:

$$F_L = F_0 + aN_m + \frac{1}{2}bN_m^2 + \cdots \tag{6}$$

with $F_0$ the Helmholtz free energy density, which is independent of $N_m$ corresponding to the normal (or mirror stable) state. Normalisation is to the ambient density $N_0$, thus attributing the dimension of energy density to the expansion coefficients $a, b$. An expansion like this one is always possible in the spirit of a perturbation approach as long as the total density $N/N_0 = 1 + N_m$ with $|N_m| < 1$. It is thus clear that $N_m$ is not the total ambient plasma density $N_0$ which is itself in pressure equilibrium with the ambient field $B_0$ under static conditions expressed by $N_0 T = B_0^2/2\mu_0$ under the assumption that no static current $\boldsymbol{J}_0$ flows in the medium. Otherwise its Lorentz force $\boldsymbol{J}_0 \times \boldsymbol{B}_0 = -T\nabla N_0$ is compensated by the pressure gradient force already in the absence of the mirror mode and includes the magnetic stresses generated by the current. This case includes a stationary contribution of the free energy $F_0$ around which the mirror state has evolved.

What concerns the presence of the mirror mode, we know that it must as well be in balance between the local plasma gradient $\nabla N_m$ of the fluctuating pressure and the induced magnetic pressure $(\delta\boldsymbol{B})^2/2\mu_0$. Note that all quantities are stationary;

the prefix $\delta$ refers to deviations from "normal" thermodynamic equilibrium, not to variations. Moreover, we have Maxwell's equations which in the stationary state reduce to

$$\nabla \times \delta\boldsymbol{B} = \mu_0 \delta\boldsymbol{J}, \quad \text{and} \quad \delta\boldsymbol{B} = \nabla \times \boldsymbol{A} \tag{7}$$

accounting for the vanishing divergence by introducing the fluctuating vector potential $\boldsymbol{A}$ (where we drop the $\delta$-prefix on the vector potential). This enables writing the kinetic part of the free energy of the particles involved in the canonical operator form

$$\frac{\boldsymbol{p}^2}{2m} = \frac{1}{2m}\Big| -i\alpha\nabla - q\boldsymbol{A} \Big|^2 \tag{8}$$

referring to ions of positive charge $q > 0$, and the constant $\alpha$ naturally has the dimension of a classical action. (There is a little problem to what is meant by the mass $m$ in this expression, to which we will briefly return below.) In this form the momentum acts on a complex dimensionless "wave function" $\psi(\boldsymbol{x})$ whose square

$$\big|\psi(\boldsymbol{x})\big|^2 = \psi^*(\boldsymbol{x})\psi(\boldsymbol{x}) = N_m \tag{9}$$

we below identify with the above used normalised excess in plasma density known to be present locally in any of the mirror mode bubbles.

Unlike quantum theory, $\psi(\boldsymbol{x})$ is not a single particle wave function, it rather applies to a larger compound of trapped particles (ions) in the mirror modes which behave similar and are bound together by some correlation length (a very important parameter, which is to be discussed later). It enters the expression for the free energy density thus providing the units of energy density to the expansion coefficients $a, b$. In the quantum case (as for instance in the theory of superconductivity) we would have $\alpha = \hbar$; in the classical case considered here, $\alpha$ remains undetermined until a connection to the mirror mode is obtained. Clearly, $\alpha \gg \hbar$ cannot be very small because the gradient and the corresponding wave vector $\boldsymbol{k}$ involved in the operation $\nabla$ are of the scale of the inverse ion gyro-radius in the mirror mode. Hence, we suspect that $\alpha \propto T/\omega_p$ where $T$ is a typical plasma temperature (in energy units), and $\omega_p$ is a typical frequency of collective ion oscillations in the plasma. Any such oscillations naturally imply the existence of correlations which bind the particles (ions) to exert a collective motion and which gives rise to the field $\boldsymbol{A}$ and density fluctuations $\delta N$. Such frequencies can be either plasma $\omega_p = \omega_i = e\sqrt{N/\epsilon_0 m}$, cyclotron $\omega_c = eB/m$ frequencies or some unknown average turbulent frequency $\langle \omega_{turb} \rangle$ on turbulence scales shorter than the typical average mirror mode scale. For the ion mirror mode the choice is that $q \propto +e$, and $m \propto m_i$.

Inspecting Eq. (8) we will run into difficulties when assuming $q = e$ and $m = m_i$ because with a large number of particles collectively participating each contributing a charge $e$ and mass $m$ the ratio $q^2/m$ will be proportional to the number of particles. In superconductivity this provides no problem because pairing of electrons tells that mass and charge just double which is compensated in Eq. (8) by $m \to 2m$. Similarly, in the case of the mirror mode we have for the normalised density excess $N_m = \delta\mathcal{N}/\mathcal{N} \equiv \zeta < 1$, where $\mathcal{N}$ is the total particle number, and $\delta\mathcal{N}$ its excess. We thus identify an effective mass $m^* \equiv \Delta m_i$, where $\Delta = 1 + \zeta$. Because of the restriction on $\zeta < 1$ this yields for the effective mass in mirror modes the preliminary range

$$m_i < m^* < 2m_i \tag{10}$$

which is similar to the mass in metallic superconductivity. However, each mirror bubble contains a different number $\delta \mathcal{N}$ of trapped particles. Hence $\zeta(\boldsymbol{x})$ becomes a function of space $\boldsymbol{x}$ which varies along the mirror chain, and $\Delta(\boldsymbol{x})$ then becomes a function of space. The restriction on $\zeta < 1$ makes this variation weak. For an observed chain of mirror modes one defines some mean effective mass $m_{eff}$ by

$$m_{eff} \equiv \langle m^*(\boldsymbol{x}) \rangle = \langle \Delta(\boldsymbol{x}) \rangle m_i \qquad (11)$$

Averaging reduces $\Delta$, making the effective mass closer to the lower bound $m_i$, which is to be used below for $m \to m_{eff}$ wherever the mass appears.

Retaining the quantum action and dividing by the charge $q$, the factor of the Nabla operator becomes $\hbar/q = \Phi_0 e/2\pi q$. Hence, $\alpha$ is proportional to the number $\nu = \Phi/\Phi_0$ of elementary flux elements in the ion-gyro cross section, which in a plasma is a large number due to the high temperature $T_\perp$. This makes $\alpha \gg \hbar$.

With these assumptions in mind we can write for the free energy density up to second order in $N_m$

$$F = F_0 + a\left|\psi\right|^2 + \frac{1}{2}b\left|\psi\right|^4 + \frac{1}{2m}\left|\left(-i\alpha\nabla - q\boldsymbol{A}\right)\psi\right|^2 + \frac{\delta\boldsymbol{B} \cdot \boldsymbol{B}_0}{2\mu_0} \qquad (12)$$

Inserted into the Gibbs free energy density, the last term is absorbed by the Gibbs potential. Applying the Hamiltonian prescription and varying the Gibbs free energy with respect to $\boldsymbol{A}$ and $\psi, \psi^*$ yields (for arbitrary variations) an equation for the "wave function" $\psi(\boldsymbol{x})$

$$\left[\frac{1}{2m}\left(-i\alpha\nabla - q\boldsymbol{A}\right)^2 + a + b\left|\psi\right|^2\right]\psi = 0 \qquad (13)$$

which is recognised as a nonlinear complex Schrödinger equation. Such equations appear in plasma physics whence waves undergo modulation instability and evolve towards the general family of solitary structures.

It is known that the nonlinear Schrödinger equation can be solved by inverse scattering methods and, under certain conditions, yields either single solitons or trains of solitary solutions. To our knowledge, the nonlinear Schrödinger equation has not yet been derived for the mirror instability because no slow wave is known which would modulate its amplitude. Whether this is possible is an open question which we will not follow up here. Hence the quantity $\alpha$ remains undetermined for the mirror mode. Instead, we chose a phenomenological approach which is suggested by the similarity of both, the mirror mode effect in ideally conducting plasma and the above obtained nonlinear Schrödinger equation to the phenomenological Landau-Ginzburg theory of metallic superconductivity.

In the thermodynamic equilibrium state the above equation does not describe the mirror mode amplitude itself. Rather it describes the evolution of the "wave function" of the compound of particles trapped in the mirror mode magnetic potential $\boldsymbol{A}$ which it modulates. This differs from superconductivity where we have pairing of particles, escape from collisions with the lattice and superfluidity of the paired particle population at low temperatures. In the ideally conducting plasma we have no collisions but, under normal conditions, also no pairing and no superconductivity though, in the presence of some particular plasma waves, attractive forces between neighbouring electrons can sometimes evolve (Treumann & Baumjohann, 2014). In superconductivity the pairing implies that the particles become correlated, an effect which in plasma must also happen whence

the superconducting mirror mode Meissner effect occurs, but it happens in a completely different way via correlating large numbers of particles, as we will exemplify farther below.

The wave function $\psi(\boldsymbol{x})$ describes only the trapped particle component which is responsible for the maintenance of the pressure equilibrium between the magnetic field and plasma. In a bounded region one must add boundary conditions to the above equation which imply that the tangential component of the magnetic field is continuous at the boundary and the normal components of the electric currents vanish at the boundary because the current has no divergence. The current, normalised to $N_0$, is then given by

$$\delta\boldsymbol{J} = \frac{iq\alpha}{2m}\left(\psi^*\nabla\psi - \psi\nabla\psi^*\right) - \frac{q^2}{m}|\psi|^2\boldsymbol{A} \tag{14}$$

which shows that the main modulated contribution to the current is provided by the last term, the product of the mirror particle density $|\psi|^2 = N_m$ times the vector potential fluctuation $\boldsymbol{A}$, which is the mutual interaction term between the density and magnetic fields. One may note that the vector potential from Maxwell's equations is directly related to the magnetic flux $\Phi$ in the wave flux tube of radius $R$ through its circumference $A = \Phi/2\pi R$.

One also observes that under certain conditions in the last expression for the current density the two gradient terms of the $\psi$ function partially cancel. Assuming $\psi = |\psi(\boldsymbol{x})|e^{-i\boldsymbol{k}\cdot\boldsymbol{x}}$ the current term becomes

$$\delta\boldsymbol{J} = \frac{q\alpha}{m}\boldsymbol{k}|\psi|^2 - \frac{q^2}{m}|\psi|^2\boldsymbol{A} \tag{15}$$

The first term is small in the long wavelength domain $k\alpha \ll 1$. Assuming that this is the case for the mirror mode, which implies that the first term is important only at the boundaries of the mirror bubbles where it comes up for the diamagnetic effect of the surface currents, the current is determined mainly by the last term which can be written

$$\delta\boldsymbol{J} \approx -\frac{q^2 N_0}{m}N_m\boldsymbol{A} = -\epsilon_0\omega_{im}^2\boldsymbol{A} \tag{16}$$

This is to be compared to $\mu_0\delta\boldsymbol{J} = -\nabla^2\boldsymbol{A}$ thus yielding the penetration depth

$$\lambda_{im}(\tau) = c/\omega_{im}(\tau) \tag{17}$$

which is the ion inertial length based on the relevant temperature dependence of the particle density $N_m(\tau)$ for the mirror mode, where we should keep in mind that the latter is normalised to $N_0$. Thus, identifying the reference temperature as $T_\perp$, we recover the connection between the mirror mode penetration depth and its dependence on temperature ratio $\tau$ from thermodynamic equilibrium theory in the long wavelength limit with main density $N_0$ constant on scales larger than the mirror mode wavelength.

## 4   Magnetic penetration scale

So far we considered only the current. Now we have to relate the above penetration depth to the plasma, the mirror mode. What we need, is the connection of the mirror mode to the nonlinear Schrödinger equation. Because treating the nonlinear

Schrödinger equation is very difficult even in two dimensions, this is done in one dimension, assuming for instance that the cross section of the mirror structures is circular with relevant dimension the radius. In the presence of a magnetic wave field $A \neq 0$ Eq. (13) under homogeneous or nearly homogeneous conditions, with the canonical gradient term neglected, has the thermodynamic equilibrium solution

$$N_m = |\psi|^2 = -\frac{a}{b} - \frac{q^2 N_0}{2mb} A^2 > 0 \tag{18}$$

which implies that either $a$ or $b$ is negative. In addition there is the trivial solution $\psi = 0$ which describes the initial stable state when no instability evolves. The Helmholtz free energy density in this state is $F = F_0$. Equation (12) tells that the thermodynamic equilibrium has free energy density

$$F = F_0 - \frac{q^2 a N_0}{2mb} A^2 - \frac{a^2}{2b} = F_0 - \frac{q^2 a N_0}{2mb} A^2 - \frac{B_{crit}^2}{2\mu_0} \tag{19}$$

where the last term is provided by the critical magnetic field which is the external magnetic field. Thus $b > 0$ and $a < 0$, and the dependence on temperature $\tau$ can be freely attributed to $a$. Comparison with Eq. (2) then yields that

$$a(\tau) = -B_{crit}^0 \sqrt{\frac{b}{\mu_0}} \tau^{-\frac{1}{2}} \left(1 - \tau\right)^{\frac{1}{2}} \tag{20}$$

At critical field one still has $A = 0$. Hence the density at critical field is

$$N_m(\tau) = \frac{|a(\tau)|}{b} = \frac{B_{crit}^0}{\sqrt{b\mu_0}} \tau^{-\frac{1}{2}} \left(1 - \tau\right)^{\frac{1}{2}} \tag{21}$$

which shows that the distortion of the density vanishes for $\tau = 1$ as it should be. This expression can be used in the magnetic penetration depth to obtain its critical temperature dependence

$$\lambda_{im}(\tau) = \left[\frac{m^2 b}{\mu_0 q^4 \left(N_0 B_{crit}^0\right)^2} \frac{\tau}{(1-\tau)}\right]^{\frac{1}{4}} \quad \text{m} \tag{22}$$

which suggests that the critical penetration depth vanishes for $\tau = 0$. However, $\tau = 0$ is excluded by the argumentation following (2) and Eq. (21), because it would imply infinite trapped densities. In principle, $\tau \geq \tau_{min}$ cannot become smaller than a

minimum value which must be determined by other methods referring to measurements of the maximum density in thermodynamic equilibrium. One should, however, keep in mind that $B_{crit}^0(\theta) \propto |\sin\theta|$ still depends on the angle $\theta$ which enters the above expressions.

The last two expressions still contain the undetermined coefficient $b$. This can be expressed through the minimum value of the anisotropy $\tau_{min}$ at maximum critical density $N_m \lesssim 1$ as

$$b = \frac{\left(B_{crit}^0\right)^2}{\mu_0} \tau_{min}^{-1} \left(1 - \tau_{min}\right) \tag{23}$$

(Note that for $N_m > 1$ the above expansion of the free energy $F$ becomes invalid. It is not expected, however, that the mirror mode allows the evolution of sharp density peaks which locally double the density.) With this expression the inertial length

becomes

$$\frac{\lambda_{im}(\tau)}{\lambda_{i0}} = \left[ \frac{\tau}{\tau_{min}} \left( \frac{1 - \tau_{min}}{1 - \tau} \right) \right]^{\frac{1}{4}} \tag{24}$$

When the mirror mode saturates away from the critical field, the magnetic fluctuation grows until it saturates as well, and one has $\boldsymbol{A} \neq 0$. The increased fractional density $N_m$ is in perpendicular pressure equilibrium with the magnetic field distortion $\delta\boldsymbol{B}$ through

$$\begin{aligned}
N_{sat} T_\perp &= \frac{1}{2\mu_0 N_0} \left( \boldsymbol{B}_0 - \delta\boldsymbol{B}_{sat} \right)^2 - \frac{B_0^2}{2\mu_0 N_0} \\
&= \frac{1}{2\mu_0 N_0} \left[ \nabla^2 (A^2) - (\nabla \boldsymbol{A})^2 \right]_{sat} - \frac{\boldsymbol{B}_0 \cdot \nabla \times \boldsymbol{A}}{\mu_0 N_0} \\
&\approx -\frac{q^2 a(\tau_{sat})}{2mb} A_{sat}^2
\end{aligned} \tag{25}$$

There is also a small local contribution from the magnetic stresses which results from the surface currents at the mirror boundaries in which only a minor part of the trapped particles is involved. This is indicated by the approximate sign.

The last two lines yield for the macroscopic penetration depth the expression (22). We thus conclude that Eq. (22) is also valid at saturation with $\tau = \tau_{sat}$. Measuring the saturation wavelength $\lambda_{sat}$ and saturation temperature anisotropy $\tau_{sat}$ determines the unknown constant $b$ through (23) with $\tau_{min}$ replaced with $\tau_{sat}$. Clearly

$$\tau_{min} \leq \tau_{sat} < 1 \tag{26}$$

as the mirror mode might saturate at temperature anisotropies larger than the permitted lowest anisotropy. Moreover, measurement of $\tau_{sat}$ at saturation, the state in which the mirror mode is actually observed, immediately yields the normalised saturation density excess $N_m(\tau_{sat})$ from Eq. (21) which then from pressure balance yields the magnetic decrease, i.e. the mirror amplitude. To some extent this completes the theory of the mirror mode in as far as is relates the density at saturation to the saturated normalised temperature anisotropy at given $T_\perp$ and determines the scale $\lambda_{im}$ and $\delta B(\tau_{sat})$.

## 5 The equivalent action $\alpha$

Since observations always refer to the final thermodynamic state, when the mirror mode is saturated, the anisotropy at saturation can be measured, and the value of the unknown constant $\alpha$ in the Schrödinger equation can also be determined. Expressed through $b$ and $\lambda_{im}$ at $\tau_{sat}$ it becomes

$$\alpha = \sqrt{2m}\lambda_{sat} = \frac{m}{q} \sqrt{\frac{b}{\mu_0 N_0 |a(\tau_{sat})|}} \tag{27}$$

What is interesting about this number is that it is much larger than the quantum of action $\hbar$ but at the same time is sufficiently small, which in retrospect justifies the neglect of the gradient term in the former section. It represents the elementary action in a mirror unstable plasma, where the characteristic length is given by the inertial scale $\alpha/\sqrt{2m} = \lambda_{sat}$ respectively maximum

of the normalised density $N_m$. One may note that $\alpha$ is not an elementary constant like $\hbar$. It depends on the critical reference temperature $T_\perp$, and it depends on $\tau$. Its constancy is understood in a thermodynamic sense.

Our argument applies when $A \neq 0$. In this case Eq. (13) reads

$$-\frac{\alpha^2}{2ma}\frac{d^2 f(x)}{dx^2} - f(1 - f^2) = 0, \quad \text{where} \quad f(x) = \frac{\psi(x)}{|\psi_\infty|} < 1 \tag{28}$$

and $|\psi|_\infty = \sqrt{N_{max}(x_\infty)}$ is given by the maximum density excess in the centre $x_\infty$ of the magnetic field decrease. Clearly this equation defines a natural scale length which is given by

$$\lambda_\alpha = \alpha / \sqrt{2m|a(T_\perp, \tau)|} \tag{29}$$

which, identifying it with $\lambda_{sat}$, yields the above expression for $\alpha$. For $x_\infty$ large the equation for $f$ can be solved asymptotically when $df/dx = 0$ for $f^2 = 1$ corresponding to a maximum in $N_m$. It is then easy to show by multiplication with $df/dx$ that

$$\frac{df(x)}{\sqrt{1 - f^2}} = \sqrt{2}\lambda_\alpha dx \tag{30}$$

which has the Landau-Ginzburg solution

$$f(x) = \tanh\left[\frac{x}{\sqrt{2}\lambda_\alpha}\right] \tag{31}$$

This implies that the excess density assumes the shape

$$N_m = N_{max} \tanh^2\left[\frac{x}{\sqrt{2}\lambda_\alpha}\right] \tag{32}$$

It approaches $N_{max}$ for $x \to x_\infty$. The above condition on the vanishing gradient of $f$ at $x_\infty$ warrants the flat shape of the excess density at maximum $(x_\infty)$ and the equally flat shape of the magnetic field in its minimum. At $x = 0$ the amplitude $f(x)$ starts increasing with finite slope $f'(0) = \sqrt{2}\lambda_\alpha$. On the other hand, the initial slope of $N_m$ is $N_m'(0) = 0$. The normalised excess density has a turning point at $x_t \approx 0.48\lambda_\alpha$ with value $N_m(x_t) \approx 0.11N_{max}$. This behaviour is schematically shown in Figure 1. Of course, these considerations apply strictly only to the one-dimensional case. It is, however, not difficult to generalise them

to the cylindrical problem with radius $r$ in place of $x$. The main qualitative properties are thereby retained. In the next section we will turn to the question of generation of chains of mirror mode bubbles, as this is the case which is usually observed in space plasmas.

Since the quantum of action enters the magnetic quantum flux element $\Phi_0 = 2\pi\hbar/e$ we may also conclude that in a mirror unstable plasma the relevant magnetic flux element is given by $\Phi_m = \alpha/q$.

Indentification of $\alpha$ is an important step. With its knowledge in mind the nonlinear Schrödinger equation for the hypothetical saturation state of the mirror mode is (up to the coefficient $b$ which, however, is defined in (23) and can be obtained from measurement) completely determined and thus ready for application of the inverse scattering procedure which solves it under any given initial conditions for the mirror mode. It thus opens up the possibility to further investigate the final evolution of the mirror mode. Executing this programme should, under various conditions, provide the different forms of the mirror mode in its

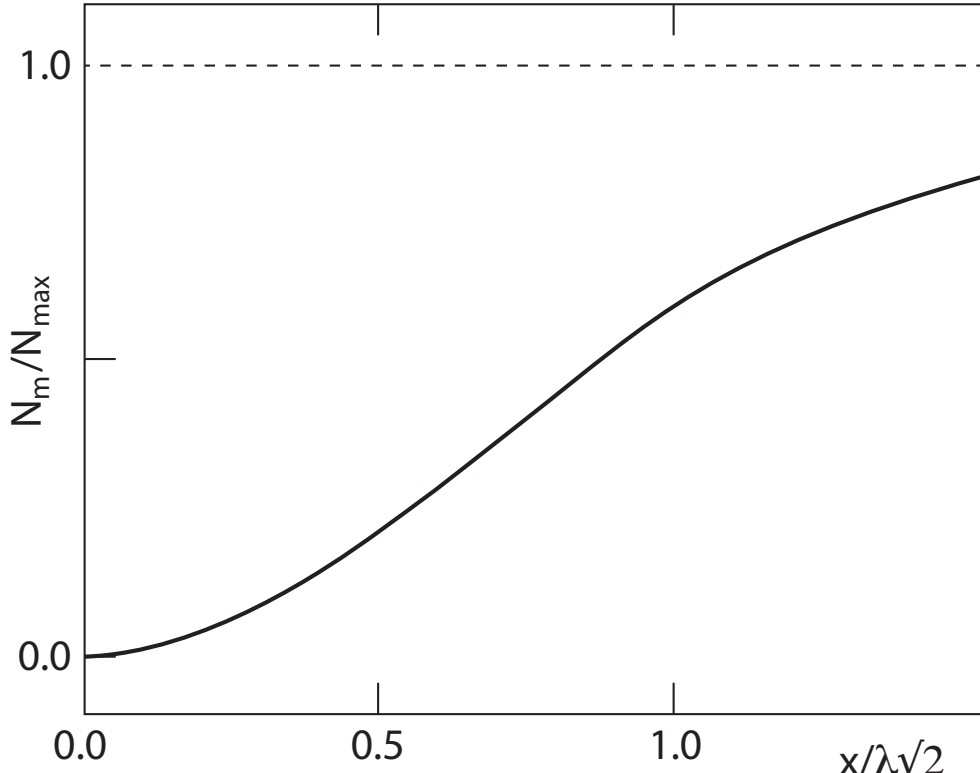

**Figure 1.** Shape of excess density in dependence on $x/\sqrt{2}\lambda_\alpha$. The shape of the magnetic field depression can be obtained directly from pressure balance. It mirrors the excess density.

final thermodynamic equilibrium state. This is left as a formally sufficiently complicated exercise which will not be treated in the present communication. Instead, we ask for the conditions under which the mirror mode evolves into a chain of separated mirror bubbles, which requires the existence of a microscopic though classical correlation length.

## 6 The problem of the correlation length

5 The present phenomenological theory of the final thermodynamic equilibrium state of the mirror mode is modelled after the phenomenological Landau-Ginzburg theory of superconductivity as presented in the cited textbook literature. From the existence of $\lambda_{im}$ we would conclude that, under mirror instability, the magnetic field inside the plasma volume should decay to a minimum value determined by the achievable minimum $\tau_{sat}$ of the temperature ratio. This conclusion would, however, be premature and contradicts observation where chains or trains (cf., e.g., Zhang et al., 2009, for examples) of mirror mode

10 fluctuations are usually observed (though also isolated "solitary mirror" modes have occasionally been reported, see e.g., Luehr & Kloecker N, 1987; Treumann et al., 1990, where they were dubbed "magnetic cavities"), which presumably are in

their saturated state having had sufficient time to evolve beyond quasilinear saturation times and reached saturation amplitudes much in excess of any predicted quasilinear level. In fact, observations of mirror modes in their growth phase have to our knowledge never yet been reported. On the other hand, in no case known to us a global reduction of the gross magnetic field in an anisotropic plasma has been identified yet.

It is clear that in any real collisionless high temperature plasma neither $N_m$ can become infinite, nor can $\tau$ drop to zero. Since it is not known how and on which way, i.e. by which exactly known process mirror modes saturate in their final thermodynamic equilibrium state, their growth must ultimately become stopped when the particle correlation length comes into play. The nature of such a correlation length is unknown, nor is it precisely defined. There are at least three types of candidates for an effective correlation length, the Debye scale $\lambda_D$, the ion gyroradius $\rho$, and some *turbulent* correlation length $\ell_{turb}$.

In a plasma the shortest *natural* correlation length is the Debye length $\lambda_D$ which under all conditions is much shorter then the above estimated penetration length $\lambda_{im}$. Referring to the Debye length, the Landau-Ginzburg parameter, i.e. the ratio of penetration to correlation lengths, in a plasma as function of $\tau$ becomes

$$\kappa_D \equiv \frac{\lambda_{im}(\tau)}{\lambda_D(\tau)} \approx \frac{c}{v_{\perp th}(\tau)} \gg 1 \tag{33}$$

a quantity that is large. Writing for the Debye length

$$\lambda_D^2(\tau) = \lambda_{D\perp}^2 \frac{1+\tau/2}{N_m(\tau)}, \qquad \lambda_{D\perp}^2 = \frac{4}{3}\frac{T_\perp/m_i}{\omega_{i0}^2} \tag{34}$$

the Landau-Ginzburg parameter can be expressed in terms of $\tau$, exhibiting only a weak dependence on the temperature ratio $\tau < 1$:

$$\kappa_D(\tau) = \frac{\lambda_{i0}}{\lambda_{D\perp}}\sqrt{\frac{2}{1+\tau/2}} \gg 1 \tag{35}$$

Thus, $\kappa_D$ is practically constant and about independent on the temperature anisotropy. Its value $\kappa_{D0} = \lambda_{i0}/\lambda_D$ at $\tau = 1, T_\parallel =$
$T_\perp$ refers to the isotropic case when no mirror instability evolves.

This is an important finding because it implies that in a plasma the case that the magnetic field would be completely expelled from the volume of the plasma cannot be realised. Different regions of extension substantially larger than $\lambda_D$ are (electrostatically) uncorrelated. They therefore behave separately lacking knowledge about their (electrostatically) uncorrelated neighbours separated from them at distances substantially exceeding $\lambda_D$. Each of them experiences the penetration scale and adjusts itself
to it. This is in complete analogy to Landau-Ginzburg theory. Thus, once the main magnetic field in an anisotropic plasma drops below threshold, the plasma will necessarily evolve into a chain of nearly unrelated mirror bubbles which interact with each other because each occupies space. In superconductivity this corresponds to a type II superconductor. Mirror unstable plasmas in this sense behave like type II superconductors. They decay into regions of normal magnetic field strength and embedded domains of spatial scale $\lambda_m(\tau)$ with reduced magnetic field. These regions contain an excess plasma population
which is in pressure and stress balance with the magnetic field. Its diamagnetism (perpendicular pressure) keeps the magnetic field partially out and causes weak diamagnetic currents to flow along the boundaries of each of the partially field-evacuated domains. This trapped plasma behaves analogously to the pair plasma in metallic superconductivity, this time however at the

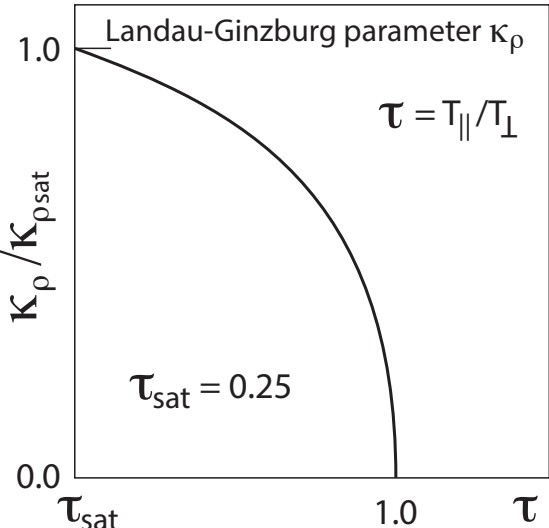

**Figure 2.** The Landau-Ginzburg parameter $\kappa_\rho/\kappa_{\rho,sat}$ as function of anisotropy ratio $\tau = T_\parallel/T_\perp < 1$ for the particular choice $\tau_{sat} = \frac{1}{4}$. The parameter $\kappa_\rho$ refers to the thermal gyroradius as the short-scale correlation length, as explained in the text. It maximises at saturation anisotropy $\tau = \tau_{sat}$ and vanishes for $\tau = 1$ when no instability sets on. For any given ratio $\tau$ the value of $\kappa_\rho$ lies on a curve like the one shown. There is a threshold for the mirror mode to evolve into bubbles which it must overcome. It is given by the ratio $\kappa_{\rho,sat} > 1$ of the critical Alfvén speed to perpendicular thermal velocity.

high plasma temperature being bound together not by pairing potentials but – in the case of the Debye length playing the role of the correlation length – by the Debye potential over the Debye correlation length.

However, the Debye length is a *very short* scale, in fact the shortest collective scale in the plasma, and though it must have an effect on the collective evolution of particles in plasmas, it should be doubted that, on the mirror mode saturation scale, it would have a substantial or even decisive effect. Instead, there could also be larger scales on which the particles are correlated.

Such a scale is, for instance, the thermal ion gyroradius $\rho(\tau)$. For the low frequencies of the mirror mode, the magnetic moment $\mu(\tau) = T_\perp/B(\tau) = \mathrm{const}$ of the particles is conserved in their dynamics, which implies that all particles with same magnetic moment $\mu(\tau)$ behave about collectively, at least in the sense of a gyro-kinetic theory.

However, though $\mu(\tau)$ is a constant of motion, it still is a function of the anisotropy through the dependence of the magnetic field on $\tau$. Expressing the thermal gyroradius through the magnetic moment

$$\rho(\tau) = \sqrt{\frac{2\mu(\tau)}{e\omega_{ci}(\tau)}} = \rho_0 \sqrt{\frac{\tau}{1-\tau}}, \qquad \rho_0 = \sqrt{\frac{2mT_\perp}{e^2\left(B_{crit}^0\right)^2}} \tag{36}$$

it can be taken as another kind of collective correlation scale as on scales *larger* than $\rho$ it collectively binds particles of same magnetic moment which, in particular, are magnetically trapped like those which are active in the mirror instability. Below the gyroradius charged particles are magnetically free. $\rho$ is the scale where the particles magnetise, start feeling the magnetic field effect and collectively enter another phase in their dynamics. This scale is much larger than the Debye length and may be more

appropriate for describing the saturated behaviour of the mirror mode. Thus one may argue that, as long as the penetration depth (inertial sale) exceeds $\rho$, the thermal gyroradius is the relevant correlation length. Only when it drops below the gyroradius, the Debye length takes over. The Landau-Ginzburg parameter then becomes

$$\kappa_\rho(\tau) = \frac{\lambda_{im}(\tau)}{\rho(\tau)} = \frac{\lambda_{i0}}{\rho_0} \left[ \frac{\tau_{sat}}{\tau} \frac{1-\tau}{1-\tau_{sat}} \right]^{\frac{1}{4}} \tag{37}$$

This ratio depends on the temperature anisotropy $\tau = T_\parallel / T_\perp$, which is a measurable quantity and the important parameter, while it saturates at $\kappa_{\rho,sat} = \lambda_{i0}/\rho_0$, the ratio of inertial length to gyroradius at critical field. This ratio is not necessarily large. It can be expressed by the ratio of Alfvén velocity $V_A$ to perpendicular ion thermal velocity $v_{\perp th}$

$$\kappa_{\rho,sat} = \frac{\lambda_{i0}}{\rho_0} = \frac{V_A\left(B^0_{crit}\right)}{v_{\perp th}} > 1 \tag{38}$$

Hence, when referring to the thermal ion gyroradius as correlation length, the mirror mode would evolve and saturate into a

chain of mirror bubbles only, when the Alfvén speed $V_A > v_{\perp th}$ exceeds the perpendicular thermal velocity of the ions. [Since $B^0_{crit} \propto \left| \sin\theta \right|$, highly oblique angles are favoured. The range of optimum angles has recently been estimated (Treumann & Baumjohann, 2018a).] This is to be multiplied with the $\tau$-dependence, of which Figure 2 gives an example. The value of this function is always smaller than one. For a chain of mirror bubbles to evolve in a plasma, the requirement $\kappa_\rho > 1$ can then be written as

$$1 \le \frac{\tau}{\tau_{sat}} < \frac{\kappa^4_{\rho,sat}}{1 + \left(\kappa^4_{\rho,sat} - 1\right)\tau_{sat}} \tag{39}$$

which is always satisfied for $\tau_{sat} < 1$ and $\kappa_{\rho,sat} > 1$, i.e. the Alfvén speed exceeding the perpendicular thermal speed, which indeed is the crucial condition for mirror modes to evolve into chains and become observable, with the gyroradius playing the role of a correlation length. Mirror mode chains in the present case are restricted to comparably cool anisotropic plasma conditions, a prediction which can be checked experimentally to decide whether or not the gyroradius serves as correlation

length.

  Otherwise, when the above condition is not satisfied and $\tau < 1$ is below threshold, a very small and thus probably not susceptible reduction in the overall magnetic field is produced in the anisotropic pressure region over distances $L \gg \rho$, much larger than the ion gyroradius. Observation of such domains of reduced magnetic field strengths under anisotropic pressure/temperature conditions would indicate the presence of a large-scale typ I classical Meissner effect in the plasma. Such

a reduction of the magnetic field would be difficult to explain otherwise and could only be understood as confinement of plasma by discontinuous boundaries of the kind of tangential discontinuities.

  The relative rarity of observations of mirror-mode chains or trains seems to support the case that the gyroradius, not the Debye length, plays the role of the correlation length in a magnetised plasma under conservation of the magnetic moments of the particles. From basic theory it cannot be decided which of the two correlation lengths, the Debye length $\lambda_D$ or the ion

gyroradius $\rho$, dominates the dynamics and saturation of the mirror mode. A decision can only be established by observations.

  However, the thermal ion gyroradius, though being the statistical average of the *distribution* of gyroscales, is itself just a plasma parameter which officially lacks the notion of a *genuine* correlation length. For this reason one would rather refer to

the third possibility, a *turbulent* correlation length $\ell_{turb}$ which evolves as the result of either high-frequency plasma or – in the case of mirror modes probably better suited – magnetic turbulence in the plasma.

It is well known that, for instance, the solar wind or the magnetosheath carry a substantial level of turbulence which mixes plasmas of various properties and obeys a particular spectrum. In the solar wind such spectra have been shown to exhibit approximate Kolmogorov-type properties, at least in certain domains of frequencies respectively wave numbers. Similarly in the magnetosheath, where the conditions are more complicated because of the boundedness of the magnetosheath and the resulting spatial confinement of the plasma and its streaming. Such spectra imply that particles and waves are not independent but bear some knowledge about their behaviour in different spatial and frequency domains, in other words they are correlated.

Unfortunately, the turbulent correlation length is imprecisely defined. No analytical expressions have been provided yet which would allow to refer to it in the above determination of the Landau-Ginzburg parameter. This inhibits predicting the range and parameter dependences of the turbulent Landau-Ginzburg ratio. Nonetheless, turbulent correlation scales might dominate the development of the mirror mode. The observation of a spectrum of mirror modes that is highly peaked around a certain wavelength not very much larger than the ion gyroradius may tell something about its nature. The above theory should open a way of relating a turbulent correlation length to the properties of a mirror unstable plasma. The condition is simply that the *turbulent* Landau-Ginzburg parameter

$$\kappa_{turb}(\tau) = \frac{\langle \lambda_{im}(\tau) \rangle}{\ell_{turb}(\tau)} > 1 \tag{40}$$

is large, depending on the anisotropy parameter $\tau$ and the average transverse scales of the mirror bubbles. This expression yields an upper limit for the turbulent correlation length

$$\ell_{turb}(\tau) < \langle \lambda_{im}(\tau) \rangle \tag{41}$$

where $\langle \lambda_{im}(\tau) \rangle$ is known as function of $\tau$ and the plasma parameters. Investigating this in further detail both observationally and theoretically should throw additional light on the nature of magnetic turbulence in high temperature plasmas like those of the solar wind and magnetosheath. It would even contribute to a more profound understanding of magnetic turbulence in general as well as in view of its application to astrophysical problems.

## 7 Conclusions

The mirror mode is a particular zero frequency mesoscale plasma instability which provides some mesoscopic structure to an anisotropic plasma. It has been observed surprisingly frequently under various conditions in space, in the solar wind, cometary environments, near other planets and, in particular behind the bow shock (Czaykowska et al., 1998) such that one also believes that they occur in shocked plasmas if the shock causes a temperature anisotropy $\tau < 1$ (cf., e.g., Balogh & Treumann, 2013, Chpt. 4). Since mirror modes are long-scale, they provide the plasma a very particular spatial texture. Mirror unstable plasmas are apparently built of a large number of magnetic bottles which contain a trapped particle population. This makes mirror modes most interesting even in magnetohydrodynamic terms as kind of a long wavelengths source of turbulence. In addition

their boundaries are surfaces which separate the bottles and thus have the character or tangential discontinuities or surfaces of diamagnetic currents which are produced by the internal interaction between the plasma and magnetic field. We have shown above that such an interaction resembles superconductivity, i.e. a classical Meissner effect.

Mirror modes in the anisotropic collisionless space plasma apparently represent a classical *thermodynamic* analogue to a "superconducting" equilibrium state. One should, however, not exaggerate this analogy. This equilibrium state is *no macroscopic quantum state*. It is a classical effect. The analogy is just formal, even though it allows to conclude about the *final mirror equilibrium*. Sometimes such an analogue helps understanding the underlying physics[1] like here where it paves the way to a global understanding of the final saturation state of the mirror mode even though this does not release us from understanding on which way this final state is dynamically achieved.

In contrast to metallic superconductivity which is described by the Landau-Ginzburg theory to which we refer here or, on the microscopic quantum level, by BCS-pairing theory, the problem of circumventing friction and resistance is of no interest in ideally conducting space plasmas which evolve towards mirror modes. High temperature plasmas are classical systems in which no pairing occurs and BCS theory is not applicable. Those plasmas are *already* ideally conducting. In contrast, there is a vital interest in the opposite problem, how a finite sufficiently large resistance can develop under conditions when collisions and friction among the particles are negligible. This is the problem of generating *anomalous* resistivity which may develop from high-frequency kinetic instabilities or turbulence and is believed to be urgently needed for instance causing dissipation in reconnection. In the zero-frequency mirror mode it is of little importance even asymptotically, in the long term thermodynamic limit, where such an anomalous resistance may contribute to decay of the mirror-surface currents which develop and flow along the boundaries of the mirror bubbles. The times when this happens are very long compared with the saturation time of the mirror instability and transition to the thermodynamic quasi-equilibrium which has been considered here.

The more interesting finding concerns the explanation *why at all*, in an ideally conducting plasma, mirror *bubbles* can evolve. Fluid and simple kinetic theory demonstrate that mirror modes occur in the presence of temperature anisotropies thereby identifying the *linear growth rate* of the instability. Trapping of large numbers of charged particles (ions, electrons) in accidentally forming magnetic bottles/traps cause the mirror instability to grow. The present theory contributes to clarification of this mechanism and its *final thermodynamic* equilibrium state as a *nonlinear* effect which is made possible by the available free energy which leads to a particular nonlinear Schrödinger equation. The *perpendicular* temperature in this theory plays the role of a critical temperature. When the parallel temperature drops below it, which means that $1 > \tau > \tau_{min}$, mirror modes can evolve. Interestingly the anisotropy is restricted from below. The parallel temperature cannot drop below a minimum value. This value is open to determination by observations.

---

[1] In a recent paper (Treumann & Baumjohann, 2018c) we have shown that a classical Higgs mechanism is responsible for bending the free space O-L and X-R electromagnetic modes in their long-wavelength range away from their straight vacuum shape when passing a plasma. The plasma in that case acts like a Higgs field and attributes a tiny mass to the photons making them heavy. This is interesting because it shows that any bosons become heavy only in permanent interaction with a Higgs field and only in a certain energy-momentum-wavelength range. It also shows that earlier attempts of measuring a permanent photon mass by observing scintillations of radiation (and also by other means) have just measured this effect. Their interpretations as upper limits for a real permanent photon mass are incorrect because they missed the action of the plasma as a classical Higgs field.

The observation of chains of mirror bubbles, for instance in the magnetosheath, which provide the mirror-unstable plasma a particular intriguing magnetic texture, suggests that the plasma, in addition to being mirror unstable, is subject to some correlation length which determines the spatial structure of the mirror texture in the saturated thermodynamic quasi-equilibrium state. This correlation length can be either taken as the Debye scale $\lambda_D$, which then naturally makes it plausible that many such mirror bubbles evolve, because in all magnetised plasmas the magnetic penetration depth by far exceeds the Debye length and makes the Landau-Ginzburg parameter based on the Debye length $\kappa_D \gg 1$. This, however, should lead to rather short scale mirror bubbles. Otherwise, the role of a correlation length could also be played by the thermal ion gyroradius $\rho$. In this case the conditions for the evolution of the mirror mode with the many observed bubbles become more subtle, because then $\kappa_\rho \gtrsim 1$ occurs under additional restrictions implying that the Alfvén speed exceeds the perpendicular thermal speed. This prediction has to be checked and possibly verified experimentally. A particular case of the dependence of the gyroradius based Landau Ginzburg parameter $\kappa_\rho$ is shown graphically in Fig. 2.

It may be noted that the Debye length and the ion gyroradius are fundamental plasma scales. Correlations can of course also be provided by other means, in particular by any form of turbulence. In that case a *turbulent* correlation length would play a similar role in the Landau-Ginzburg parameter, whether shorter or larger than the above identified penetration scale. What concerns mirror modes in the magnetosheath to which we referred (Treumann & Baumjohann, 2018a), it is well known that the magnetosheath hosts a broad turbulence spectrum in the magnetic field as well as in the dynamics of the plasma (fluctuations in the velocity and density).

Though this makes it highly probable that turbulence intervenes and affects the evolution of mirror modes, any "turbulent correlation length" is, unfortunately, rather imprecisely defined as some average quantity. To our knowledge, though when referring to multi-spacecraft missions not impossible, it has even not yet been precisely identified in any observations of turbulence in space plasmas. Even when identified, its functional dependence on temperature and density is required for application in our theory. If these functional dependencies are not available, it becomes difficult to include any turbulent correlation length. In addition, one expects that its turbulent nature would make the theory nonlocal. Attempts in that direction must, at this stage of the investigation, be relegated to future efforts.

Finally, it should be noted that the magnetic penetration depth $\lambda_m$ which lies at the centre of our investigation is rather different from the ordinary inertial length scale of the plasma. It is based on the excess density $N_m < 1$ less than the bulk plasma density $N_0$. It thus gives rise to a enhanced (excess) plasma frequency $\omega_m = \omega_i \sqrt{N_m + 1} = \omega_i \sqrt{1+\zeta} \lesssim \sqrt{2}\omega_i$ which implies that $L > c/\omega_i > \lambda_m$ is shorter than the typical scale of the volume $L$ and (slightly) shorter than the bulk inertial length $c/\omega_i$. This becomes clear when recognising that the mirror mode evolves inside the plasma from some thermal fluctuation (cf., Yoon & López, 2017, for the calculation of low-frequency thermal magnetic fluctuation levels in a stable isotropic plasma; similar calculations in stable anisotropic plasmas have not yet been performed) which causes the magnetic field locally to drop below its critical value Eq. (2). Then $\lambda_m$ identifies the local perpendicular scale of a mirror bubble after it has saturated and is in thermodynamic equilibrium. One expects that the transverse diameter of a single mirror bubble in the ideal case would be roughly $2\lambda_m$. However, since each bubble occupies *real* space, in a mirror saturated plasma state the bubbles compete for space and distort each other (cf., e.g., Treumann & Baumjohann, 1997, for a sketch) thereby providing the plasma an

*irregular magnetic texture* of some, probably narrow, spectrum of transverse scales which peaks around some typical transverse wavelength and resembles a strongly distorted crystal lattice that is elongated along the ambient magnetic field.

It also relates the measurable saturated magnetic amplitudes of mirror modes to the saturated anisotropy $\tau_{sat}$ and the Landau-Ginzburg parameter $\kappa$, transforming both into experimentally accessible quantities. These should be of use in the development of a weak-kinetic turbulence theory of magnetic mirror modes as the result of which mirror modes can grow to the observed large amplitudes which are known to by far exceed the simple quasilinear saturation limits. It also paves the way to the determination of a (possibly turbulent) correlation length in mirror unstable plasmas of that so far no measurements have been provided.

To the space plasma physicist the present investigation may look a bit academic. However, it provides some physical understanding of how mirror modes do really saturate, why they assume such large amplitudes, evolve into chains of many bubbles or magnetic holes, and what the conditions are that this happens. Moreover, since the mirror mode to some sense resembles superconductivity, which also implies that some population of particles involved behave like a superfluid, so it would be of interest to infer whether such a population exhibits properties of a superfluid. One suggestion is that the untrapped ions and electrons which escape from the magnetic bottles along the magnetic field resemble such a superfluid population. This also suggests that other high-temperature plasma effects like the formation of purely electrostatic electron holes in beam-plasma interaction may exhibit superfluid properties. In conclusion, the unexpected working of the thermodynamic treatment in the special case of the magnetic mirror mode shows once more the enormous explanatory power of thermodynamics.

*Acknowledgement.* This work was part of a Visiting Scientist Programme at the International Space Science Institute Bern. We acknowledge the interest of the ISSI directorate and the hospitality of the ISSI staff. We thank the referees K.-H. Glassmeier (TU Braunschweig, DE), O.D. Constantinescu (U Bukarest, RO), and H.-R. Mueller (Dartmouth College, Hanover, NH, USA) for their critical remarks on the manuscript, in particular the non-trivial questions of the effective mass and the role of turbulence.

*Data availability.* No data sets were used in this article.

*Competing interests.* The authors declare that they have no conflict of interest.

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
