# Peer review of "The mirror mode: A "superconducting" space plasma analogue"

_Annales Geophysicae, 2018_

## Referee Comment (RC1) · K.-H. Glassmeier (Referee) · 15 May 2018

I read this paper with largest interest! The attempt to relate a classical plasma physics problem to well established quantum physics in a structurally comparable form is most interesting! Actually, the authors provide a most valuable contribution as they also suggest future experimental studies. This study should be published as it is.
* * *

---

## Author Comment (AC1) · 26 May 2018

We thank the referee for his encouraging report.

It is indeed surprising though satisfactory to find that a similarity exists between quantum and classical effects. This should not be exaggerated, because it is (so far) just a similarity and does not mean that the mirror mode is a macroscopic quantum effect.

We would like to highlight at this place some points (which in the manuscript have not been discussed as they are more general than the mere mirror instability).

The analogue circumvents stepping up the ladder from linear to fully nonlinear theory of the mirror mode which in purely classical thinking is rather difficult if not impossible. The reference to quantum methods enables to directly go to the final thermodynamic

equilibrium state even though the result describes just the saturated state of the mirror mode but does not illuminate how and in which way this state is reached. (Trying to do this in classical thermodynamics, i.e. without reference to a "wave function", turned out to run into unsurmountable difficulties, which explains why so far no steady state other than quasilinear has been investigated for mirror modes.) This is the same as in superconductivity which is based on stationary quantum theory while not beeing an evolution theory in time. It just distinguished two phases: the normal (resistive) and the abnormal (superconducting) phases, which works because of the pairing of electrons and bose-einstein condensation of these pairs in the lowest energy state where they become superfluid and escape resistance.

This kind of pairing is clearly absent in a plasma with temperatures by far exceeding any pair binding energies. However, there is a similar kind of "pairing" viz. trapping of large numbers of particles when magnetic bottles form by chance and thermal anisotropy allows the number of trapped particles to grow. Those particles can be considered as forming one large quasiparticle. But in contrast to superconductivity there is no change in mass because the ratio of charge to mass remains constant and spins play no role for two reasons: the plasma is classical and, in addition, the average spin averaged over all directions of the large number of trapped particles is zero plus possibly one which plays no role (recall that pairs have twice the mass and twice the charge, but in their pairing interaction the doubling of mass in the spin interaction term does not drop out but causes their bosonic nature). The trapped particles therefore are statistically all bosons, behave like bosons and allow the description by one common wave function.

The untrapped particle population are the particles of high energy respectively those with large parallel velocity. They escape trapping.

This we did in the mean field mean wave function approach where the ratio (according to the above discussion) $N_m q / N_m m = q/m$ is the same as for one particle. One can take this as justification of proof for the validity of our approach.

[Figure]

It would be interesting to investigate the superfluid behaviour of this trapped population. It will give rise to three types of waves: ion-acoustic waves (purely electrostatic), Alfven waves (purely electromagnetic), electromagnetic ion-cyclotron waves (no linear dispersion). Hence the former if sufficiently short wavelength play the role of phonons in superfluidity, i.e. not acting like a viscosity/resistance, the latter are like rotons. Hence, near criticality the trapped particles behave like a spuerfluid.

This behaviour should be taken into account in any more precise theory. This means that one should refer to the Bose distribution with finite mass and temperature anisotropy and variable ratio of trapped to ambient plasma density, consider Alfven waves (liner dispersion) as photons and ion-cyclotron waves as rotons. Then one would be in the position to develop a more precise thermodynamic theory of the mirror modes.

In summary, this is an interesting concept.

Once more many thanks to the reviewer.

––––––––––––––––––––––––––

---

## Referee Comment (RC2) · D. Constantinescu (Referee) · 3 Jul 2018

The manuscript explores the similarity between mirror mode unstable space plasmas and superconductive metals. Based on this analogy, a parallel phenomenological formalism is constructed for the saturated mirror mode at thermodynamic equilibrium. In the proposed interpretation, the role of the quantum electron pairs in the Landau-Ginzburg theory of superconducting metals is taken in space plasmas by collectives of ions moving coherently over the correlation length. The Meissner effect in metals is paralleled by the onset of mirror instability which causes local depletions of the magnetic field in space plasmas. The critical parameter controlling the system is now the orthogonal temperature in the anisotropic space plasma.

[Figure]

A link to the experimental data is the prediction of the proposed theory that - if the relevant correlation length is the ion gyroradius - mirror mode chains form only when the Alfvén speed exceeds the perpendicular thermal speed.

A further development suggested by the authors is solving the equivalent nonlinear Schrödinger equation which would provide the solutions for the saturated mirror mode at thermodynamic equilibrium. Achieving this would be a significant result in the physics of mirror modes.

Minor points:

page 4

- lines 9-11: One would expect the ambient density $N_0$ to be perturbed both ways (i.e. increase where the ambient field decreases, and decrease where the ambient field increases). Then the normalised density $N_m$ would be negative in the higher field regions. In this case the inequality $N_m < 1$ should read $|N_m| < 1$.

- equation (8): Since $\psi$ applies to a compound of correlated ions, what represents the mass $m$ and charge $q$ in eq. (8)? a single particle or the whole compound?

page 5

- line 11: should the inequality be $<$ rather then $>$ ?

In the reviewer's opinion, the manuscript should accepted for publication.

---

## Author Comment (AC2) · 3 Jul 2018

Thank you very much for the kind and constructive review and criticisms, detection of the typo ($>$ instead of $<$) and the suggestion to include the absolute signes in $|N_m|$.

In particular, however, thanks very much for alerting us concerning the real value of the mass $m$. Indeed, we have skipped the discussion of this problem which may become quite sensible in other cases because it is not in general solvable. In the mirror mode it can however be solved similar to solid state superconductivity.

In metallic superconductivity, as you rightly remark, pairing requires introducing $m* = 2m$ for the mass because the mass doubles due to pairing. It is thus known and remains uncompensated by the double charge. This is the similar in the mirror bubbles with the

important difference that there is of course no pairing and the absolute number of trapped particles $\delta\mathcal{N}$ which contribute to pressure balance is not known and changes from bubble to bubble. Only the density excess $N_m$ is known.

However, this is a normalised quantity. We know that $|N_m| = |\delta\mathcal{N}|/\mathcal{N} = \zeta < 1$. Hence the relevant mass becomes $m^* = (1 + \zeta)m_i = \Delta m_i$. Again, however, this is a function of space $\vec{x}$ since it is different from bubble to bubble.

On the other hand, one may define an *average* equivalent mass $m = \langle m^*(\vec{x}) \rangle$ averaged over the entire length of the mirror chain. This then becomes $m_i < m < 2m_i$ and is the mass which is to be understood in all following equations.

We are very grateful to this reviewer for insisting on us to include this clarification.

We will include a short clarifying remark on this item in the MS when resubmitting. So far we gently went by this problem as we thought it might cause more complications than clarifications when applied to other instabilities where the additional constraint $|N_m| < 1$ is lacking. Nevertheless, we agree with this reviewer that it is quite reasonable to point this difficulty out clearly also at this place and show the way how to include the modification of $m^*$ in mirror modes. The practical realisation of this programme still remains problematic.

We feel that by imposing thermodynamic considerations of this kind in discussing the final quasi-equilibrium state of an instability like the mirror mode makes very much sense since it is in praxi not feasible to calculate the full chain of interactions in weak or strong turbulence theory. The many contributing modes and particle populations can hardly be defined. Thermodynamics as the global theory gives instead a clue on the final equilibrium state. It works perfectly in superconductivity. That it also works, if only approximately, also in the mirror mode is very satisfactory though not that much surprising.

Once more many thanks for this most constructive review and the suggestions.

With best wishes Wolfgang and Rudolf

---

## Author Comment (AC3) · 4 Jul 2018

In our reply we mentioned that for the large number of trapped ions in anyone of the mirror bubbles the ratio $q/m$ of charge and mass remains unaffected even when many particles are trapped and the mirror mode evolves along the indicated lines until reaching its thermodynamic final equilibrium state. This is doubtlessly true.

However Reviewer 2 noted that in some places the mass occurs without reference to charge, and there the "effective mass" might assume a different value. This is also true. We have been aware of that problem but did not want to mention it for not unnecessarily overloading the anyway complicated theory. We answered the problem affirmatively in very simple terms (see our reply to R2).

[Figure]

To make it short: Indeed the effective mass of the mirror-trapped ions changes when all the excess ions behave like one correlated object (in superconductivity the mass doubles when Cooper pairs form). In mirrors no pairs form, thus the mass increases by the ratio of excess number of particles to total particle number $\delta\mathcal{N}/\mathcal{N} = \zeta$, where $\zeta < 1$. Hence the effective mass of the particles which contribute to the thermodynamic state of the mirror mode becomes $m^* = (1+\zeta)m_i$. This value is bounded as $m_i < m^* < 2m_i$ which is very similar to superconductivity.

In real mirror modes the number of trapped particles varies from bubble to bubble. The value $\zeta(\vec{x})$ depends on space and must be averaged over the total chain of mirror modes. The effective mass then becomes $m_{eff} \equiv \langle m^*(\vec{x}) \rangle = m_i(1 + \langle \zeta(\vec{x}) \rangle)$. The above range of the effective mass remains unaffected.

---

## Referee Comment (RC3) · H.-R. Müller (Referee) · 13 Jul 2018

This is a very interesting and well-formulated paper that addresses the phenomenology of the saturation of mirror mode evolution, and its manifestation as a chain of mirror mode bubbles. The treatment abandons the typical quasilinear analysis with its limited usefulness. A detailed comparison of the theoretical description of mirror mode plasma physics with that of type II superconductors described macroscopically by Landau-Ginzburg theory is laid out successfully. The investigation yields testable predictions, and if measurements confirm the findings, the comparative analysis will expand our cursory knowledge of mirror mode evolution and saturation. This is a valuable treatment of mirror mode plasma physics, as a more comprehensive description based on the physics of (weak) turbulence seems intractable.

[Figure]

This paper is written well, and by necessity the analysis is intriguingly interdisciplinary, meticulously pointing out the distinction between quantum and classical phenomena in the correspondence analysis. I have no comments to add to this paper, and provide only a list of minor typos.

page 2 line 31: capital theta_i: the index should be j here. page 6 line 20: typo page 7 line 6: full stop needed. page 9 line 23: Identification page 10 line 26: adjusts page 11 line 7: longer -> larger; also some instances further below. page 12 line 26: type
* * *

---

## Author Comment (AC4) · 13 Jul 2018

We cordially thank this reviewer for his very constructive comments and suggestions. We will, of course, take them into account in any resubmission.

In addition we will, to the information of this (as well as the other) reviewer(s) include a reference to a possible "turbulent" correlation length which we had not considered yet, because no treatable expression of it is know to us. Possibly a turbulent correlation length is more appropriate and also more important than the other two (Debye radius or gyroradius). The magnetosheath (where mirror modes are permanently observed behind the shock) is known to contain a highly turbulent plasma.

The observation of a particular "saturation length scale" of mirror modes should in this

case put an upper imit on the turbulent correlation length. This might be of general importance in mhd turbulence theory.

Once more many thanks to this reviewer for his encouraging comments.